# Curcumin Formulations and Trials: What’s New in Neurological Diseases

**DOI:** 10.3390/molecules25225389

**Published:** 2020-11-18

**Authors:** Stella Gagliardi, Carlo Morasso, Polychronis Stivaktakis, Cecilia Pandini, Veronica Tinelli, Aristides Tsatsakis, Davide Prosperi, Miriam Hickey, Fabio Corsi, Cristina Cereda

**Affiliations:** 1Genomic and Post Genomic Unit, IRCCS Mondino Foundation, 27100 Pavia, Italy; stella.gagliardi@mondino.it (S.G.); cecilia.pandini@mondino.it (C.P.); 2Istituti Clinici Scientifici Maugeri IRCCS, 27100 Pavia, Italy; carlo.morasso@icsmaugeri.it (C.M.); veronicatinelli16@gmail.com (V.T.); davide.prosperi@unimib.it (D.P.); fabio.corsi@icsmaugeri.it (F.C.); 3Medical School, University of Crete, 70013 Heraklion, Greece; polychronis.stivaktakis@gmail.com (P.S.); tsatsaka@uoc.gr (A.T.); 4Department of Biology and Biotechnology “L. Spallanzani”, University of Pavia, 27100 Pavia, Italy; 5NanoBioLab, Dipartimento di Biotecnologie e Bioscienze, Università di Milano-Bicocca, 20126 Milano, Italy; 6Department of Pharmacology, Institute of Biomedicine and Translational Medicine, University of Tartu, 50411 Tartu, Estonia; miriam@ut.ee; 7Department of Biomedical and Clinical Sciences “Luigi Sacco”, University of Milan, 20157 Milano, Italy

**Keywords:** curcumin, neurological diseases, nanoformulation, clinical trials

## Abstract

Curcumin’s pharmacological properties and its possible benefits for neurological diseases and dementia have been much debated. In vitro experiments show that curcumin modulates several key physiological pathways of importance for neurology. However, in vivo studies have not always matched expectations. Thus, improved formulations of curcumin are emerging as powerful tools in overcoming the bioavailability and stability limitations of curcumin. New studies in animal models and recent double-blinded, placebo-controlled clinical trials using some of these new formulations are finally beginning to show that curcumin could be used for the treatment of cognitive decline. Ultimately, this work could ease the burden caused by a group of diseases that are becoming a global emergency because of the unprecedented growth in the number of people aged 65 and over worldwide. In this review, we discuss curcumin’s main mechanisms of action and also data from in vivo experiments on the effects of curcumin on cognitive decline.

## 1. Introduction: The Need to Protect Curcumin in the Body

First isolated in 1815 by Vogel and Pelletier, curcumin’s chemical structure was determined by Lampe e Milobedeska in 1913. It is a diferuloylmethane polyphenol characterized by two phenolic rings (as with resveratrol), an active component of turmeric and it is extracted from the rhizome of the tropical plant Curcuma longa (family Zingiberaceae). It has a long history of use as a dietary agent and food preservative. Curcumin has been used in Ayurvedic medicine for thousands of years and is eaten by millions on a daily basis. It is classified as generally recognised as safe (GRAS) by the Food and Drug Administration (FDA) and is available over the counter, as a supplement, in many countries.

With antioxidant [1], anti-inflammatory and antitumor properties [2,3], curcumin has received much attention in several neurodegenerative diseases, such as Alzheimer’s (AD), Huntington (HD) and Parkinson’s diseases (PD) [4,5]. Extensive research over several decades has sought to identify the mechanisms of molecular action of curcumin [6]. It regulates inflammatory cytokines, growth factors, growth factor receptors, enzymes, adhesion molecules, proteins related to apoptosis and cell-cycle proteins, such as cyclin, and modulates its molecular targets by altering their gene expression, signaling pathways or through direct interaction. We discuss these pharmacodynamics in more detail below.

We must also note the pharmacokinetics of curcumin. Curcumin is lipophilic, which is beneficial for absorption because it will aid in penetrating lipid membranes. However, this increases the difficulty of its formulation as a medicinal product. By far, the most convenient route of administration of medicines is oral, for subsequent enteral absorption. Following oral administration, ingested curcumin passes through the stomach, where practically no absorption takes place [7], and it is then detected in the intestine [8,9]. However, curcumin is then metabolized extensively during first-pass metabolism [10], which results in low bioavailability. The human liver and gut microsomes treated with 20 μM curcumin form glucuronides of curcumin and its phase I metabolites [11]. It is very difficult to model human metabolism accurately in animals, but rat mucosa metabolises curcumin [12] and the major metabolites in male rats gavaged with curcumin (Meriva; 340 mg/kg (curcumin) and euthanized 20 min afterwards) were also glucuronides of curcumin and desmethoxycurcumin, with some evidence for reduction [13]. Even if administered intraperitoneally, lipophilic substances, such as curcumin, will drain into the hepatic portal vein for transport to liver and metabolism, prior to distribution around the body [14]. Indeed, following ip administration of 0.1 g/kg curcumin to mice, glucuronides were also the major metabolites [15]. Thus, both gut and liver metabolise curcumin with Phase II processes being important in several species [16]. In the rat, these metabolites are excreted in bile and faeces [17,18]. However, the glucuronidated metabolites may be subject to enterohepatic recirculation [19].

Once in the gut, curcumin may change the composition of bacterial populations and reduce intestinal inflammation [20,21]. Interestingly, perturbations in gut microbiota have been associated with different pathologies, such as neurological diseases (AD, PD) [22,23]. Curcumin, by reducing inflammation, may modulate the gut-brain axis and act as a neuroprotective factor [24]. This may be an important beneficial effect of curcumin in addition to its effects after absorption, where it can modulate numerous signalling pathways [18]. Furthermore, many of curcumin’s metabolites are bioactive [25,26] and so efforts to enhance the initial bioavailability of the parent compound (curcumin) could also lead eventually to increased levels of these secondary beneficial molecules.

We and others have shown nanomolar levels of curcumin in brain following oral administration of unmodified curcumin to mice [26,27]. However, improving curcumin’s bioavailability is not a trivial task, and careful analysis of effects in brain and on behaviour are required for new formulations because micromolar levels may be deleterious due to curcumin’s well-known capacity to inhibit the proteasome. Moreover, it has been reported that the combination of spirulina-curcumin-Boswellia is effective in reducing the size of benign thyroid nodules and can be safely administered to patients. The mechanism of the reduction of the nodules is attributed to the anti-inflammatory effects of the curcumin [28].

Many different methods of nanoformulation of curcumin have been attempted with the aim of improving brain levels in the context of neurodegenerative diseases and improving aqueous solubility (Table 1). As an example, early studies demonstrated a large increase in bioavailability of orally administered curcumin in plasma [29,30] and bioavailability of ip- or iv-administered curcumin in brain [31,32] when it was encapsulated within PLGA nanoparticles (poly(lactide-co-glycolide) nanoparticles) and compared with native curcumin. PLGA is considered safe as it is broken down in vivo into glycolic acid and lactic acid, which are themselves utilised in the TCA cycle for ultimate excretion as H_2_O and CO_2_. The addition of groups such as polyethylene glycol protects PLGA particles from premature degradation via the reticulo endothelial system [33] and increases the plasma bioavailability of curcumin even further following oral administration [34]. Loading curcumin onto human serum albumin nanoparticles increased plasma and peripheral tissue levels following iv administration [35], an approach that took advantage of ex vivo findings showing that serum albumin is transported across endothelia [36]. Pellets of solid dispersion of curcumin using polyvinylpyrrolidone also increased plasma levels compared with conventional curcumin following oral administration [37], and other authors showed that optimised nanostructured lipid carriers were superior to solid lipid nanoparticles for increasing and preserving curcumin levels in brain (iv administration) [38]. Polymeric nanoparticles (NanoCurc) composed of N-isopropylacrylamide, vinylpyrrolidone and acrylic acid also resulted in measurable levels of curcumin in mouse brain (i.p. injection) [39]. It has been also shown that the amphiphilic PVP-OD is capable of forming nanocarriers in aqueous media, as well as solubilizing highly hydrophobic compounds, such as curcumin. Two commonly used methods for drug loading, namely emulsification and ultrasonic dispersion, lead to the formation of carriers with different particle size distributions and, more importantly, a marked biological effect as antiinflammation [40].

In this review, we describe how curcumin may have an important role to play in neurological diseases and we also discuss the evolution of curcumin trials and formulations.

## 2. Curcumin and Neurological Disease: Focusing on Mechanism

The possibility of using curcumin as a treatment for neurological diseases is a relatively recent development. Although the specific effects may differ for each disease, we can identify common pathways by which curcumin exerts its effects. For instance, inflammation is a common factor in many neurological diseases. In most cases, NFκB plays a pivotal role in the inflammatory response by inducing TNF-α, which in turn mediates ROS activation. Curcumin has an inhibitory effect on NFκB, thereby reducing inflammation [45]. Neurological diseases, such as multiple sclerosis (MS), spinal cord injury (SCI), and AD [46], show immune cell invasion and another mechanism that can be modulated by curcumin is T-lymphocyte activation and monocyte chemotaxis. Curcumin negatively regulates pathways that modulate chemokines responsible for monocyte recruitment and activation of T-lymphocytes [47,48].

### 2.1. Alzheimer’s Disease

The presence of amyloid or plaques comprising β-amyloid (Aβ) peptide is one the major characteristics of AD. Curcumin acts on Aβ levels, deposits, and aggregation in a multitude of ways. In vitro studies demonstrated that compounds derived from curcumin protect PC12 cells against Aβ(25–35) and Aβ(1–42) insult [49]. When used as a pretreatment, curcumin prevents Aβ-induced toxicity in the human SH-SY5Y neuroblastoma cell line by increasing mRNA and protein levels of mitochondrial genes, such as nuclear respiratory factor 1 (Nrf1) and mitochondrial transcription factor A (TFAM) [50]. Bisdemethoxycurcumin (BDC) pretreatment in an ex vivo study on peripheral blood mononuclear cells (PBMCs) of AD patients showed that curcumin decreases mRNA levels of NFκB, BACE1 and Toll-like receptor and upregulates mRNA levels of mannosyl-glycoprotein 4-β-*N*-acetylglucosaminyltransferase (MGAT3) and vitamin D receptor (VDR), leading to diminished Aβ aggregates [51]. This protective effect of curcumin has been confirmed also in vivo studies. In a mouse model of AD, curcumin decreased the levels of inflammatory molecules, such as IL1β, TNF-α, cyclooxygenase 2 (COX 2) and nitric oxide (NO), while also decreasing the number of glial fibrillary acidic protein-(GFAP) and Iba-1-positive cells, thus suppressing the neuroinflammatory response [52]. Moreover, Curcumin was associated with a protective effect on some forms of memory: although no effects were noted on spatial memory in the object location test and Y maze in a rat model of AD treated with 50 or 100 mg/kg curcumin, it did prevent deficits in recognition memory in the object recognition test [53].

Finally, curcumin can also protect against Tau-induced neurotoxicity by inhibiting glycogen synthase kinase-3 (GSK-3), which regulates the phosphorylation of Tau protein [54].

### 2.2. Parkinson’s Disease

In vitro studies and animal models have demonstrated that curcumin is protective in PD via different mechanisms, such as inhibition of apoptosis, oxidative stress, and Lewy body accumulation.

Treatment with Demethoxycurcumin (DMC), which is an additional curcuminoid present in turmeric, protected SH-SY5Y cells against rotenone-induced cytotoxicity by increasing cell viability: this effect was likely due to a reduction in reactive oxygen species and restoration of the imbalanced expression profile of Bax, BAD, caspase-3, caspase-6, caspase-8, and caspase-9 in mitochondria and Cyt-c in cytosol, all of which were increased by rotenone, and the expressions of Bcl-2, Bcl-xL, and Cyt-c in mitochondria, which were decreased significantly following rotenone treatment [55]. Another mechanism observed in SH-SY5Y cells is that the level of heat shock protein 90 (HSP90) was increased following MPTP treatment, which was reversed by curcumin. Interestingly, HSP90 expression is greatly increased in PD patient brain and its levels correlate with increased levels of insoluble α-Synuclein (α-Syn) [56]. Importantly, HSP90 silencing significantly reduced curcumin’s effect, as cell proliferation and inhibiting apoptosis, while its overexpression increased the influence of curcumin on PD [57].

In a rat PD model (2,4,5-trihydroxyphenethylamine (6-OHDA)-induced), curcumin reduced the death of dopaminergic neurons in substantia nigra and it inhibited astrocyte activation, most likely through the Wnt/β-Catenin signaling pathway. Canonical Wnt signaling is mediated mainly by β-Catenin protein, which is a co-activator of transcription factors including lymphoid enhancer factor (LEF) and T-cell factor (TCF) [58]. Accumulating evidence over the last decade showed that Wnt/β-Catenin signaling, in partnership with glial cells, is critically involved in nigrostriatal dopaminergic neuronal health, protection, and regeneration, suggesting that Wnt/β-Catenin signaling could boost a full neurorestorative program in PD [59]. Moreover, curcumin ameliorated behavioral disturbances (rotational behavior) in this model [60,61]. In the rotenone PD rat model, curcumin exerts neuroprotective effects by ameliorating motor deficits and increasing antioxidant enzyme activities [62]. Curcumin also protected against rotenone-induced excitatory tetanic potentiation in hippocampus [63]. Moreover, curcumin can bind α-Syn directly in the hydrophobic non-amyloid-β region, and inhibit formation of oligomers and fibrils [64,65]. Recently, Liu et al. developed an intranasal delivery system for the brain-targeted delivery of a curcumin analogue-based nanoscavenger (curcumin analogue molecules with polyethylene glycol). The authors showed that this treatment stimulated nuclear translocation of transcription factor EB, one of the major autophagy regulators, thereby triggering both autophagy and calcium-dependent exosome secretion for the clearance of α-Syn [66]. Finally, chronic and acute curcumin treatment of transgenic mice overexpressing GFP-tagged wild-type α-Syn improved motor behavior, which was correlated with an increase in phosphorylated α-Syn protein in the cortical presynaptic terminals [67].

### 2.3. Amyotrophic Lateral Sclerosis

It has been demonstrated that curcumin regulates the early aggregation phases of reduced superoxide dismutase 1 (SOD1), one of the main proteins associated with Amyotrophic Lateral Sclerosis (ALS), leading to the production of non-fibrillar smaller and less toxic aggregates [66]. It also binds these products strongly, blocking their aggregation sites and, thus, inhibits the production of toxic species [68]. The role of curcumin treatment in ALS has been observed also using NSC-34 cells transfected with the DNA-RNA binding protein TDP-43. TDP-43 is one of the major disease proteins in the pathological inclusions of ALS, and it has been associated with both gain- and loss-of-function consequences including altered over responsiveness to cellular stressors, increases in DNA damage, splicing regulation and transcriptome-wide changes [69]. DMC rescues the impairment induced by the overexpression of TDP-43 by exerting a protective effect on mitochondrial membrane potential and decreasing the levels of mitochondrial uncoupling protein 2 (UCP2) protein [70], which has neuroprotective effects by modulating mitochondrial transmembrane potential, mitochondrial proliferation, ATP and ROS levels [71]. Curcumin also influences the initiation and propagation of action potentials, which are enhanced by the overexpression of wild-type and mutant Q331K TDP-43; indeed, abnormal action potentials and voltage-gated sodium channels on motoneuron-like cell lines was significantly ameliorated following DMC treatment, which may act by alleviating oxidative stress and mitochondrial dysfunction [72]. Recently, a randomized pilot clinical trial demonstrated that nanocurcumin is safe and may improve survival when used as an add-on treatment in patients with ALS, especially patients with existing bulbar symptoms [73].

### 2.4. Multiple Sclerosis

Curcumin can contribute to the restoration of myelination and causes an anti-inflammatory response by acting on astrocytes in the central nervous system. In particular, in a cellular model of MS comprising U373-MG cells (human glioblastoma astrocytoma cell line) treated with lipopolysaccharide (LPS), curcumin reduced both the release of IL6 and MMP9 activity [74]. Similar studies were also conducted in the autoimmune encephalomyelitis (EAE) model for MS where curcumin inhibited the expression of IFNγ, IL17, and IL12 cytokines in the central nervous system [75]. In a rat MS model, curcumin increased the level of myelination by potentiating protective pathways against oxidative stress, the Nrf2 pathway, by restoring iNOS mRNA levels and by enhancing brain-derived neurotrophic factor (BDNF), nerve growth factor (NGF), platelet-derived growth factor receptor α (PDGFR α), nestin (a neural stem cell marker), myelin basic protein (MBP), Olig2, and oligodendrocyte progenitor markers [76]. Also, in vivo and in vitro experiments conducted with curcumin nanoformulations (dendrosomal nanoparticles, DNC) demonstrated that DNC increased oligodendrogenesis from subventricular zone-derived neural stem cells and oligodendrocyte progenitor cells in a dose-dependent manner, and furthermore, enhanced remyelination activity of transplanted neural stem cells by promoting their survival and oligodendrogenesis capacity [77].

### 2.5. Spinal Cord Injury

Studies on the effects of curcumin in SCI have been conducted mostly on rodent models with different types of damage. Using a balloon compression method, it has been demonstrated that curcumin treatment improved locomotor and sensory performance in the first week after SCI [78]. It also reduced inflammatory cell invasion and NFkB activity and decreased glial scar development, which is a severe obstacle to neuronal regeneration [78].

In a hemi-section model of SCI in rats, curcumin downregulated GFAP expression thereby blocking astrocyte activation and, thus, reduced neurological deficit [79].

Anti-inflammatory effects of curcumin were observed also in a contusion mouse model and in an ischemia-reperfusion injury rat model where curcumin reduced NO, TNFα and IL1β tissue protein. Moreover, curcumin reduced caspase-3 levels after injury [80].

In a clip-compression model, curcumin plays an anti-inflammatory role by attenuating both TNFα and NFkB [81]. It inhibits glial scar formation via different mechanisms: suppression of NFkB activity and limitation of GFAP [82] and downregulation of chemokine release by astrocytes. Finally, in this model, curcumin up-regulates the protein expression of HO-1, which protects cells against oxidative injury [81], and up-regulates zonula occludens-1 protein (ZO-1), thus limiting the SCI-induced disruption of tight junctions and maintaining blood-spinal cord barrier integrity [81].

In an animal study of traumatic spinal cord injury in Sprague Dawley rats, transplantation of olfactory ensheathing cells treated with curcumin promoted neural regeneration and functional recovery. Curcumin led to the production of anti-inflammatory and neurotrophic factors and the reduction in pro-inflammatory cytokines [83]. Finally, in human astrocytes and the white matter injury model of SCI, curcumin inhibited hydrogen peroxide (H_2_O_2_)-induced oxidative stress, reduced hypoxia-induced expression of HIF-1, GFAP, and NF-H proteins, and inhibited apoptosis [84].

### 2.6. Stroke

Curcumin can ameliorate stroke pathology in different ways. In the stroke-prone spontaneously hypertensive rat model, it has been demonstrated that curcumin can have a protective effect by delaying the onset of stroke and increasing the probability of survival, most likely because of a curcumin-induced physiological regulation of mitochondrial ROS generation [85]. These data were confirmed in a HUVECs cellular model, where curcumin treatment attenuated H_2_O_2_-induced oxidative stress [81]. Similar results were obtained from other animal and cellular models: curcumin treatment reduced cerebral infarction and neuronal apoptosis in a mouse model of middle cerebral artery occlusion (MCAO), and in N2a cells, it reduced mitochondrial dysfunction [86]. Moreover, using the MCAO rat model, it was observed that curcumin reduced infarct volume [87] and brain oedema at 24 h [88], prevented the development of post-ischemic neurodegeneration [89], and treated rats achieved better neurological scores compared to untreated rats [90,91]. These effects were probably due to the reduction of oxidative stress and neuronal apoptosis by increasing the mitochondrial levels of the antiapoptotic Bcl2 protein, by reducing the cytosolic translocation of Cyt-c [87] and by reducing the mitochondrial membrane potential [90]. Finally, curcumin can act also during the reperfusion phase, where it reduces vascular endothelium activation and reduces adherent neutrophils at the vascular endothelium level by lowering NFκB-mediated TNFα expression [88].

Marques et al. compared the protective effects of free versus nanoemulsified curcumin after intracerebral hemorrhage induced by collagenase in Wistar rats and demonstrated that the curcumin nanoformulation was able to improve behavioral recovery and modulate antioxidant responses, reduce the size of the haematoma and attenuate the weight loss in treated rats [92]. In contrast, free curcumin, at the same dose, failed to be effective in the same parameters [93].

## 3. Nanoformulated Curcumin: Focusing on Alzheimer’s Disease In Vivo Models

### 3.1. Diagnostic Tool for Amyloid

Native curcumin and curcumin-encapsulated nanoliposomes label plaques in 5xFAD [94] transgenic mice, a model of Alzheimer’s disease, when used as a staining tool ex vivo or when administered to mice intraperitoneally. Indeed, in this mouse model, curcumin and this nanoformulated curcumin performed similarly to antibody staining with 6E10, which is a purified anti-β-amyloid antibody reactive to aa 1–16 Aβ and to APP, and the curcumin formulations were found to be more sensitive than thioflavin S or congo red. The majority of extracellular and vascular amyloid (using 6F3D antibody) labelling was also labelled with curcumin-decorated nanoliposomes in human AD cortex [95]. Curcumin-conjugated magnetic nanoparticles of superparamagnetic iron oxide labelled plaques that were observed using MRI in Tg2576 [96] mice following iv administration [97]. Curcumin derivatives have also been developed as FRET probes (Förster resonance energy transfer probes) to quantify monomers and high-molecular-weight amyloid aggregates in solution, which could help screen aggregation inhibitors [98]. One of these probes was shown to enter brain when administered iv, and fluorescence in vivo imaging revealed that it remained for a significantly longer period of time in Tg2576 brain compared with WT mice, suggesting that it had successfully labeled amyloid in Tg2576 mice [99,100]. Zinc induces oligomerisation of amyloid β, and, similar to results in mammals, curcumin fluorescence clearly separated ZnCl_2_-induced Alzheimer flies (Drosophila) from control flies following oral administration of curcumin-conjugated carbon-dot nanoparticles [101]. Interestingly, Aβ42 from AD patient plasma was bound and sequestered by liposomes functionalised with phosphatidic acid or with a modified Apolipoprotein E-derived peptide or with a curcumin derivative, suggesting a possible future use of the liposomes as a “peripheral sink” [102]. Thus, empiric evidence shows the ability of curcumin to label and bind amyloid in several different models and in AD patient samples, and nanoformulations do not inhibit this capacity. Indeed, in addition to encapsulating curcumin, some authors have taken advantage of curcumin’s attraction to amyloid and have decorated liposomes with curcumin or curcumin derivatives [102,103] for improved targeting to amyloid. Other authors have loaded selenium PLGA nanoparticles with curcumin for improved targeting [104]. Furthermore, some have included curcumin as an additional therapeutic. For example, it was encapsulated in nanoparticles designed to diagnose cerebral amyloid angiopathy [105].

### 3.2. Therapeutic Tool for Alzheimer’s Disease

Several review articles have been published with guidelines on appropriate readouts and group size for preclinical studies on AD. Unfortunately, AD preclinical research is plagued by poor study design including a lack of blinding and randomisation and no sample size calculation [106,107,108,109,110] despite calls for these controls [111]. With these caveats, several authors have examined nanoformulated curcumin in different animal models of AD and promising beneficial effects have been noted.

Meng et al. created a rat model of AD using high-dose D-galactose and intrahippocampal injection of Aβ42. An LDL-mimic nanostructured lipid carrier modified with lactoferrin and encapsulating curcumin showed greater entry into brain and it reduced oxidative stress and neurodegeneration in the AD rats compared with free curcumin (30% (*w/v*) PEG-400 in a 5% (*v/v*) glucose solution) or curcumin loaded in unmodified NLCs (N = 10 per group, agents administered iv) [41]. Maiti et al. showed that after only five days of intraperitoneal administration of solid lipid curcumin particles, cortical and hippocampal plaques were reduced in old 5xFAD mice. These mice 5xFAD are a well-characterised transgenic mouse model of AD, and in this study, they were used at one year of age, when pathology is well established (N = 4 per group). The number of degenerating neurons in hippocampus and cortex were also reduced, as were markers of inflammation and gliosis. The effects of nanoparticulated curcumin was greater than that of free curcumin (dissolved in methanol and diluted in 0.1 M PBS for administration), which was borne out by the greater inhibition of fibrilisation and oligomerisation of Aβ42 by the nanoparticulated formulation [42]. These data indicate a benefit of treatment for diagnosed patients, who already have large levels of amyloid in brain [112].

AD is a clinically diagnosed disease; thus, an examination of the benefit of nanoformulated curcumin on behavioural readouts is critical. Several authors have examined behaviour in chemically induced models, following acute exposure to amyloid, or in genetic mouse models. As AD patients show extensive neurodegeneration, cognitive deficits, and brain amyloid load at diagnosis, both acute and chronic models will be invaluable in the search for disease-modifying drugs. However, it is likely that genetic mouse models mimic the chronic nature of AD more faithfully.

To this end, Kakkar et al. treated male Lacca mice (N = 10 per group) orally with AlCl_3_ for 18 weeks, which developed spatial learning deficits in the Morris water maze by 12 weeks, although severe toxicity was reported with only 3–4 mice surviving within each group. Therapeutic treatment was initiated at 12 weeks and continued for a further six weeks. At that point, spatial learning deficits in the Morris water maze were ameliorated in a dose-dependent manner by solid lipid nanoparticles (SLC) of curcumin (oral, 1–50 mg/kg). The performance of the SLC-curcumin nanoparticles was similar to rivastigmine, which is used in the clinic to treat AD-related cognitive impairment and was superior to free curcumin (oral, dissolved in 25% tween 80). Biochemical measurements revealed elevated lipid peroxidation in the AD mice, which was reduced by the SLC-curcumin. Here, no effect of free curcumin was observed. However, we do note that, in this particular model, aluminum is thought to increase the activity of AChE, which is unlikely to be clinically relevant. Nevertheless, SLC-curcumin reduced the activity of the enzyme in a dose-dependent manner with no effect of free curcumin [43].

In another acute model of AD, male ICR mice were subjected to intrahippocampal injections of okadaic acid (a phosphatase inhibitor). Subsequently, groups of 20 mice were treated intravenously with free curcumin or nanoparticles for 10 days (solubilisation not described; data for behaviour of N = 5 per group provided, only). The nanoparticles were human serum albumin nanoparticles loaded with curcumin and bearing T807, a novel tau positron emission tomography imaging agent, and triphenylphosphine, which targets mitochondria, that were attached to a red blood cell membrane surface. Several different combinations of nanoparticles were examined, and the curcumin-T807-TPP-RBC nanoparticles provided the greatest beneficial effect on spatial learning deficits, followed by curcumin-loaded T807-RBC nanoparticles. A similar pattern was observed on outcome measures of lipid peroxidation and on gliosis and microgliosis (N = 5 per group). Free curcumin was of no benefit [113]. In a second paper from this group, curcumin was loaded onto red blood cell membrane-coated PLGA particles carrying T807 molecules. Okadaic acid was again used to create an acute model of AD (N = 20 per group), manifesting as spatial learning deficits in the Morris water maze. Several different combinations of iv injected nanoparticle were examined but only the nanoparticles bearing curcumin, RBC membrane and the T807 ligand ameliorated these spatial learning deficits, gliosis and microgliosis while both the nanoparticles carrying curcumin and RBC membrane and the nanoparticles bearing curcumin, RBC membrane and the T807 ligand protected against hippocampal neurodegeneration [114]. These latter data emphasise the importance of examination of neuropathology with behaviour in preclinical trials and that inflammation is critical to address in the context of AD.

Fidelis et al. injected Aβ25–35 intracerebroventricularly into Swiss male mice (N = 7–8 per group), which were then gavaged orally once every second day with free curcumin in granola oil, nanocapsules loaded with curcumin or blank nanocapsules. Compared to control ICV injections, Aβ25–35 induced depressive-like behaviour in the forced swim task and Porsolt swim task, which was ameliorated by curcumin-loaded nanocapsules but not free curcumin. General activity levels were comparable across groups. Although not displayed by a majority of AD patients, depression is an important symptom in up to half of patients. SOD and catalase activities were also reduced by the curcumin-loaded nanocapsules [115].

Following intracerebroventricular injection of Aβ1–42, aged (18–22 m) female Swiss Webster mice were gavaged orally for 14 days with 1 or 10 mg/kg curcumin loaded-lipid core nanocapsules, 50 mg/kg free curcumin or saline (frequency of dosing not specified). The authors reported improved performance in the Morris water maze in free curcumin-treated mice and in mice administered with curcumin in nanoparticle form (N = 8 per group). Both of these treatments reduced the mRNA and protein expression of several inflammatory cytokines, including NFkappaB, in hippocampus, frontal cortex and serum [116].

Importantly, several authors have examined behavioural readouts following chronic treatment of established genetic mouse models of AD. In an early study, Cheng et al. [117] loaded curcumin into polyethyleneglycol-polylactide di-block polymer micelles. Although stable and protective against metabolism of curcumin in vivo, no overall beneficial effects were observed in Tg2576 mice (N = 20 per mixed sex group) treated with 23 mg/kg per week by oral gavage for three months from nine months of age [117]. Plaques typically develop by approximately 11–13 months in this model [96]. In other work by Ma et al., transgenic mice expressing wild-type humanTau under control of the tau promoter were fed 500 ppm solid lipid nanoparticles of curcumin in chow. Mice were treated from 15–16 months to 19–20 months in order to more specifically model AD patients at diagnosis since at this age extensive tau pathology and neuronal loss is observed in this model. Hippocampal heat shock proteins levels were normalized by curcumin nanoparticles and the treatment also ameliorated the redistribution of synaptic proteins observed in control-treated transgenic mice, for example, the treatment redistributed PSD-95 levels from SDS hippocampal fractions to soluble hippocampal fractions. Curcumin reduced specifically soluble dimers of tau without affecting insoluble forms. Curcumin appeared to improve learning in the treated transgenic mice because although performance of control transgenics did not improve over time in the hidden-platform version of the Morris water maze, performance of treated transgenics was identical to that of wildtype animals (N = 7–10). Nevertheless, the slower swim speeds of the control transgenic mice may have impacted their performance, and deficits in control transgenic mice were not observed in the Y maze or novel object recognition task, precluding the measurement of a treatment effect in these additional memory tasks.

APP/PS1 mice develop plaques at a much earlier age than Tg2576 mice (already at 2 m [118]). Using small group sizes (N = 5 per group), Huang et al. [116], examined PLGA nanoparticles in male APP/PS1 mice. Eight-month-old mice were treated once every two days intraperitoneally for three weeks with PLGA nanoparticles of varying characteristics. Nanoparticles loaded with Aβ generation inhibitor S1 (PQVGHL peptide) and curcumin and conjugated to CRT peptide (cyclic CRTIGPSVC peptide), which targets transferrin receptor to aid in crossing the blood brain barrier, were most consistently successful in this model in addition to nanoparticles loaded with S1 and curcumin. Cognitive deficits in the two-trial Y maze and novel object recognition test were ameliorated by these two treatments and the two different treatments also reduced levels of soluble and insoluble Aβ 40 and 42. Similar to results from acute models (discussed above), behavioural improvement was associated not only with reduced plaque load but these two different treatments also reduced astrogliosis and protected against synapse loss. Using the same mouse model (APP/PS1), Fan et al. [119], injected intraperitoneally nine-month-old male mice (N = 8 per group) with PLGA-PEG nanoparticles loaded with curcumin and conjugated to a separate peptide that targets transferrin receptor (B6 peptide (CGHKAKGPRK)). All treatment groups, comprising free curcumin, PLGA-PEG-curcumin and PLGA-PEG-curcumin-B6, showed improved spatial learning, but the greatest effect was observed in mice treated with the targeted nanoparticles. All treatment groups reduced levels of plaque in the brain and reduced phosphorylated tau, again, with the targeted nanoparticles having the greatest effect.

Thus, curcumin-loaded nanoparticles are beneficial in several different models of AD, and curcumin may also contribute therapeutic effects when incorporated into multifunctional nanoparticles. Moreover, nanoformulation greatly reduces the required dose, for example 25 mg/mg once weekly for 12 weeks [119] or 2 mg/kg every two days for three weeks [117]. However, careful examination of disease models, including both behaviour and neuropathology, is required to understand the disease modifying effects of nanoparticles. This in turn necessitates appropriate timepoint selection, appropriate endpoint measurement, and appropriate group sizes to provide maximum sensitivity to detect treatment effects (Table 2).

## 4. Curcumin: Focusing on Formulations and Cognitive Skills in Human Trials

While the literature on the effects of curcumin on cognition and dementia in in-vitro and animal models is extremely detailed, studies on its effects in humans are much more limited and their interpretation is more complex.

The original idea that curcumin might have a protective role against dementia is linked to the low prevalence of this disease observed in India. The prevalence of dementia in the US is estimated to be approximatively 10% of the population over 65 years. In contrast, studies on the prevalence in an Indian rural community [120] and in the urban population of Kerala [121] suggested a much lower prevalence of 0.5% and 3.3% respectively. The consumption of high doses of curcumin in the Indian population might be just one of the reasons behind the huge difference in the prevalence of dementia, and other factors are likely to have an important role [122]. However, these initial observations sparkled some more focused studies on the topic (Table 3).

In 2005, Ng and coworkers published the first paper that explicitly focused on the effect of curcumin on cognition. In an epidemiological study on an elderly Singaporean cohort, they compared the results obtained from the Mini-Mental State Examination test (MMSE) in subjects self-reporting rare, occasional, or frequent consumption of curry. Subjects with occasional or frequent consumption of curry performed better than those reporting rare consumption, and the effect was more pronounced in men and subjects with Indian ethnicity [123].

This appealing result was the basis for the first randomized double-blind, placebo-controlled trial on the effect of curcumin in subjects affected by AD, published in 2008 by Baum and coworkers [124]. The study was conducted on 34 subjects who were treated for six months with different doses of oral curcumin (up to 4 g a day) or with placebo. Despite expectations, the results of an MMSE performed at the beginning and at the end of the treatment did not highlight any specific effect of curcumin on cognition. This was mostly due to the lack of cognitive decline in the control group, thus suggesting that curcumin is probably not curative but that it could still have some preventative effects. Additionally, the plasma levels of curcumin appeared to be extremely low as the drug was present as a glucuronidated adduct. Ringman and co-workers then published a second clinical trial on AD patients in 2012 [125]. To solve the low bioavailability of curcumin, in this trial, the subjects were treated with Curcumin C3 Complex^®^, a formulation of Curcumin with Piperine to stimulate its absorption [126]. Thirty subjects with mild to moderate AD were included and divided into three groups who were treated for 24 weeks with 2 g a day or 4 g a day of curcumin, or with placebo. Similar to previous observations, the bioavailability of curcumin remained very low and the drug was detected in plasma with the vast majority being a glucuronidated adduct. In this trial, the scores from cognitive tests performed on the participants (MMSE and ADAS-Cog) showed a slightly greater decline in curcumin groups than control, although no statistical differences were obtained.

Even if the low number of subjects included in these two studies did not allow a conclusive result, no other clinical trial has been performed on patients with AD as their cognitive function is likely already too compromised to respond to treatment with curcumin. Thus, a need for new formulations with higher bioavailability has emerged.

The first attempt to overcome these initial failures was reported in 2015 by Cox and co-workers [127]. This study focused on short-term effects of curcumin in a healthy, elderly population. To solve the bioavailability issue, subjects were treated with a solid lipid emulsion of curcumin named Longvida^®^ that was previously proven to be more effective in delivering native curcumin to plasma [128]. Sixty subjects were included in the study and a set of cognitive tests was administered after 1 h, after 3 h, and after chronic treatment for four weeks. The results showed that both short term and chronic treatment with curcumin improved working memory and digit vigilance. Very recently, the same authors performed a replication study and extended the chronic treatment up to 12 weeks on 85 subjects [129]. In this replication study, the cognitive performance of the group treated with curcumin at four weeks did not reach a statistically significant improvement. However, after 12 weeks, the subjects treated with curcumin performed significantly better on a virtual version of the Morris Water Maze, the non-virtual version of which is commonly used to assess spatial and learning memory in rodents.

Rainey-Smith and coworkers conducted a similar study in 2016 but they focused on long-term treatment [130]. Ninety-six healthy subjects were treated for 12 months with placebo or with a lipidic micro formulation of Curcumin named Biocurcumax™, which was previously reported to improve bioavailability eight-fold [131]. The cognitive performances of the participants were assessed using the Montreal Cognitive Assessment (MOCa) test. In this study, after six months of treatment, the placebo group manifested signs of cognitive decline that were not present in the group treated with curcumin, suggesting a potential protective effect. However, after 12 months, no statistically significant difference was found between the two groups as the performance of the placebo group rebounded to the baseline level.

The latest study on the potential effect of curcumin on cognitive skills was reported in 2018 by Small and coworkers [44]. This double-blind placebo-controlled clinical trial was designed, similar to Cox et al. and Rainey-Smith et al., to examine the effects of treatment in forty healthy elderly volunteers. However, in this case, the focus was on a long-term treatment of 18 months. Curcumin was administered as Theracurcumin, a nanoformulation made of purified curcumin and polysaccharides [132]. The use of this particular formulation improved the bioavailability of curcumin drastically and it was detected at a concentration of 40 μg/mL in the plasma of the treated subjects.

**Table 3 molecules-25-05389-t003:** This table summarises the main results from clinical trials of curcumin. We include the study number to allow access to original sources.

Study	Year	Target	Treatment	Effect on Cognition	Study Number
Baum et al. [124]	2008	34 Alzheimer	Curcumin	No	NCT00164749
Ringman et al. [125]	2012	30 mild Alzheimer	Curcumin C3 Complex^®^	No	NCT00099710
Cox et al. [127]	2015	60 healthy adults	Longvida^®^	Yes	ACTRN12612001027808
Rainey-Smith et al. [130]	2016	96 healthy elderly	Biocurcumax^®^	Limited	ACTRN12611000437965
Small et al. [44]	2018	40 healthy adults	Theracurmin^®^	Yes	NCT01383161

Results from cognitive skills testing (assessed using the Buschke Selective Reminding Test) were particularly interesting. In the subjects treated with curcumin, an improvement was observed already after six months and was maintained for the entire eighteen months of the trial. In contrast, the performance of the placebo group remained constant throughout. In a subgroup of subjects, the authors were also able to prove that curcumin induced a significant reduction in plaque and tangle deposition, as measured by PET, thus providing the first clinical evidence that curcumin could enhance some cognitive skills and might be beneficial in the prevention of dementia.

Overall, studies examining the effectiveness of Curcumin in dementia are limited, and the published articles do not provide a definitive answer. The use of new formulations, and in particular nano-formulations, however, seems to be a very promising approach able to enhance the bioavailability and thus the pharmacological activity of curcumin. On the other hand, each study made use of a different set of tests to measure the cognitive skills of the enrolled subjects, thus limiting the possibility to compare the results obtained. The use of a more precise and standardized set of tests in upcoming trials would be beneficial for the development of curcumin-based therapies active on cognition.

## 5. Conclusions

In conclusion, studies on curcumin and neurological diseases have shown that this natural product has important antioxidant and anti-inflammatory proprieties; it is a significant regulator of apoptosis and cell cycle and also modulates numerous molecular targets by altering their gene expression, signaling pathways or through direct interaction.

Curcumin has been object of drug experimentation in different neurodegenerative diseases but it’s very low bioavailability hinders its use in oral formulations. Thus, a need for new formulations with higher bioavailability has emerged. Several nano-formulations are capable of enhancing curcumin bioavailability, and data from several animal studies and from human trials suggest that nanoformulated curcumin could be of great benefit in AD.

The data thus far collected show that curcumin is a safe and beneficial nutraceutical product, but do not provide a conclusive result; now, new clinical trials will be necessary to evaluate completely the curcumin potential in term of dose, administration, and formulation.

## Figures and Tables

**Table 1 molecules-25-05389-t001:** Comparison of curcumin levels in brain or plasma when administered to animals as free agent or in nanoformulated form.

Type of Nanoformulation	Admin. Route	Brain/Plasma	Effect	Measured Effect	Ref.
PLGA	IV (20 mg/kg)	Brain	Increment	Free Curcumin 500 µg/g (wet tissue)	[32]
				Nano Curcumin 1400 µg/g (wet tissue)	
PVP	Oral (60 mg/kg)	Plasma	Increment	Free Curcumin 0.7 ng/mL	[37]
				Nano Curcumin 109 ng/mL	
Copolymer	IV (25 mg/kg)	Brain	Increment	Free Curcumin not tested	[39]
				Nano Curcumin 0.322 µg/g	
(LDL)-mimic Nanoparticles -Lactoferrin	IV (10 mg/kg)	Plasma	Increment	Free Curcumin not detected	[41]
				Nano Curcumin 13.03 ng/mL	
Solid Lipid Nanoparticles	IV (50 mg/kg)	Brain	Increment	Increased association of nanoformulated Curcumin with amyloids plagues. No quantitative data	[42]
Red blood cells camouflaged—Albumin NPs	IV (5 mg/kg)	Brain	Increment	Free Curcumin 0.05% of inj. Cur./g brain	[43]
				Nano Curcumin 0.25% of inj. Cur./g brain	
Theracurcumin^®^	Oral (300 mg/kg)	Plasma	Increment	Free Curcumin 0 ng/mL	[44]
				Nano Curcumin 1600 ng/mL	

**Table 2 molecules-25-05389-t002:** Summary of available studies on NP curcumin in animal models of AD examining both cognitive behavior and pathology. ↑ and ↓ arrows are used to show whether outcome measure tended towards control (normal) levels or away from control (normal) levels and are relative. -, no effect. Blank = not measured.

Treatments	Preclinical Trial in AD Model						Citation
	Behavioural Impairment: ↓ towards Normal Levels or ↑ Away from Normal Levels			Neuropathology: ↓ Towards Normal Levels or ↑ Away from Normal Levels			
	Morris water maze			Brain oxidative stress marker ^1^		Degenerating neurons	Kakkar et al., 2011
Solid lipid nanoparticles containing curcumin	↓↓			↓		↓ ^2^	
Curcumin	↓			-		- ^2^	
	Morris water maze			Astrocytes, Microglia ^3^		Hf neuronal loss	Gao, Wang et al., 2020
Red blood cell-membrane-camouflaged human serum albumin nanoparticles loaded with curcumin and bearing T807 and triphenylphosphine	↓↓			↓		↓	
RBC-membrane-camouflaged HSA NPs loaded with curcumin and bearing T807	-			-		-	
RBC-membrane-camouflaged HSA NPs loaded with curcumin and bearing triphenylphosphine	-			-		-	
RBC-membrane-camouflaged HSA NPs loaded with curcumin	-			-		-	
Curcumin	-			-		-	
	Morris water maze			Astrocytes, Microglia ^3^		Hf neuronal loss	Gao, Chu et al., 2020
Red blood cell membrane-coated PLGA particles carrying T807 molecules and loaded with curcumin	↓			↓		↓↓	
RBC membrane-coated PLGA particles loaded with curcumin	-			-		↓	
RBC membrane-coated PLGA particles carrying T807 molecules	-			-		-	
Curcumin	-			-		-	
	Morris water maze			Inflammatory cytokine expression in brain			Giacomeli et al., 2019
Lipid-core nanocapsules loaded with curcumin	↓			↓			
Curcumin	↓			↓			
	Radial arm maze		Contextual fear conditioning		Plaque area		Cheng et al., 2013
Polyethyleneglycol-polylactide di-block polymer micelles loaded with curcumin	-		-		-		
Curcumin	-		-		↓		
	Morris water maze				Tau dimers	Hf synaptic protein abnormal distribution	Ma et al., 2013
Solid lipid nanoparticle curcumin (Longvida)	↓				↓	↓	
		Two-trial Y maze	Novel object recognition	Astrocytes, Microglia ^3^	Plaque area	Hf synaptic number ^4^	Huang et al., 2017
Poly (lactide-co-glycolide) nanoparticles conjugated with cyclic CRTIGPSVC peptide and loaded with curcumin and Aβ generation inhibitor S1		↓↓	↓	↓↓	↓↓	↓	
PLGA nanoparticles loaded with curcumin and Aβ generation inhibitor S1		↓	↓	↓↓	↓↓	↓	
PLGA nanoparticles loaded with curcumin		↓	-	↓	-	-	
PLGA nanoparticles loaded with Aβ generation inhibitor S1		↓	-	-	↓	-	
	Morris water maze				Plaque burden		Fan et al., 2018
Poly (lactide-co-glycolide) nanoparticles conjugated with B6 peptide and loaded with curcumin	↓				↓		
PLGA nanoparticles loaded with curcumin	↓				↓		
Curcumin	↓				↓		

Hf, hippocampal formation; IHC, immunohistochemistry; NOR, novel object recognition; PFC, prefrontal cortex. ^1^ MDA (malondialdehyde); ^2^ Qualitative; ^3^ immunohistochemistry, ^4^ PSD95 immunohistochemistry.

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
