# Peer review of "Curcumin Formulations and Trials: What’s New in Neurological Diseases"

_molecules, 2020, doi:10.3390/molecules25225389_

Round 1
Reviewer 1 Report
The manuscript by Gagliardi et al. entitled “Curcumin formulations and trials: what’s new in neurological diseases” tries to depict the state of research of curcumin nanoformulations for the treatment of neurological diseases. However, the references to newly published studies are limited. The authors did not provide an accurate account of the current research.
- The authors need to clarify in the entire manuscript if the studies mentioned were done in vitro or in vivo.
- On page 2-lines 45-47. The authors mentioned that “curcumin has been the object of drug experimentation in different neurodegenerative diseases”. Could the authors cite the original studies and not reviews? It is essential to highlight the recent works and not the recent reviews.
- On page 2, line 47, could the authors explain the meaning of “pharmacokinetic compartment”?
- The following sentence also needs some adjustments: Moreover, curcumin is difficulty heavily metabolized [7]”, please revise.
- In the same sentence, “because of first-pass metabolism (metabolism in gut mucosa and liver), minimal amounts get into systemic blood for transport to brain following oral (enteral) administration”.
It is unnecessary to explain the first-pass metabolism; also, please describe the meaning of “systemic blood” and “systemic blood for transport to brain following oral (enteral) administration”.
- The authors mention that “Great efforts are now being made to protect curcumin from first-pass metabolism”, the authors are invited to consult the literature to realize that this effort is lasting for 40 years now. Please amend your sentence to give a realistic overview of curcumin nanomedicine.
- The authors are explicit enough for the readers to understand their meaning without the need for words between parenthesis at the end of many sentences. Please remove them along with the text and/or rephrase to be more concise.
- The authors mentioned the following: “Many different methods of nanoformulation of curcumin have been attempted with the aim of improving brain levels in the context of neurodegenerative diseases and improving aqueous solubility [11]”. Could the authors provide references mentioning the brain accumulation of curcumin and the use of these formulations to treat neurodegenerative diseases?
- Mentioning a single example for a review is not sufficient (page 2, lines 61-67). Could the authors provide a table that will regroup all the curcumin nanoformulations, the concentration in the brain achieved by these nanoparticles, and the neurodegenerative disease studied?
- The authors mentioned that “Loading curcumin onto human serum albumin nanoparticles, thought to aid in transport across endothelia, increased plasma and peripheral bioavailability [18]”. Could the authors explain that the transport across the endothelia was performed in vitro only?
- Could the authors mention that the study using NanoCurc was conducted in mice?
- The reference 29 mentioned by the authors reported the effect of the fractionation of Curcuma longa on protecting PC12 cells from beta-amyloid insult. This study did not show that “curcumin inhibits the formation of fibrillar Aβ and destabilizes fibrillar Aβ40 and Aβ42, showing a protective effect in response to Aβ toxicity. Could the authors provide a relevant reference?
- The reference 30 used by the authors is a review that is not mentioning the effect of curcumin on the expression of Nrf1or mitochondrial transcription factor in SH-SY5Y. As indicated before, it is essential to use original work and not review and provide a reference relevant to the statement made. Please rectify the manuscript with the righteous references.
- The authors mentioned that: “Moreover, it was demonstrated in a rat model of AD that curcumin can reduce the deficits in the object recognition test by increasing adult neurogenesis and decreasing neuroinflammation [34]. However, the reference 34 mentions the following: Curcumin treatment in the doses of 50 and 100 mg/kg prevented the deficits in recognition memory in the ORT, but not in spatial memory in the OLT and Y maze. Curcumin treatment exerted only slight improvements in neuroinflammation, resulting in no improvements in hippocampal and subventricular neurogenesis. Could the authors revise their statement in accordance with the reference 34 conclusion?
- The authors reported the effect of DMC “against rotenone-induced cytotoxicity by increasing cell viability” in SH-SY5Y cells. Please explain if the authors consider the effect of the pure curcumin or the mixture of curcuminoids present in turmeric. What is the effect of curcumin on the expression of Bcl2, Bad, Bax, BclXL, CytC, and caspases?
- Could the authors explain the role of HSP90 in the progression of Parkinson’s disease (page 3, lines 117-121)?
- Could the authors provide more information on the contribution of the Wnt pathway in the onset and/or progression of Parkinson’s disease? The reference 38 shows the effect of curcumin on proteins associated with cell proliferation, including β-catenin.
- The authors casually mentioned on page 3, line 124: “and improved rotational behaviour”. Please put this information into the perspective of behavioral assessment of Parkinson’s disease.
- Please explain how curcumin reversed the synaptic alteration in the hippocampus, the reference 41 only mentioned protection (page 3, line 127).
- Please provide the full name of the protein α-Syn (page 3, line 127).
- The authors wrote that “Moreover, curcumin can bind α-Syn, the main component of Lewy bodies, and hinders its accumulation in dopaminergic neurons [42]”. However, reference 42 only demonstrated the inhibition of the aggregation of α-synuclein aggregation using fibril formation and soluble oligomer formation. None of the work was conducted using dopaminergic neurons. Please correct your statement according to the finding of the publication cited as 42. Could the authors account for the latest work regarding curcumin and α-synuclein? More than a thousand publications were published since 2019 on this topic.
- Could the authors provide a reference for the first sentence of Amyotrophic lateral sclerosis (page 3, lines 133-135)?
- Could the authors explain the contribution of the TDP-43 and UCP2 proteins in the progression of ALS? The authors provided references from 2012 and 2014, while literature reported earlier work in 2020. It is critical that the authors provide the latest work in this review.
- When appropriate, could the authors mention if the work was conducted in vitro or in vivo? As an example, the following statement is not clear: “Curcumin also influences the initiation and propagation of action potentials, which are enhanced by the overexpression of TDP-43 [46]”.
- The cell line U373-MG cells described by the authors (page 4, line 146) is a human glioblastoma astrocytoma cell line. Could the authors correct their statement? It is also not clear why the authors are not providing the latest work on curcumin and multiple sclerosis. In Pubmed, 31 publications are listed since 2016, on google scholar more than 5,000 publications since 2016.
- In section 2.5. spinal cord, could the authors provide the latest information regarding curcumin? Additional works have been performed since 2016.
- The paragraph on stroke needs to account for the latest publications. Also, the authors provided two publications from 2018; more recent studies are available.
- Why did the authors not mention curcumin nanomedicines in the treatment of neurological diseases? The increased bioavailability is critical for curcumin to reach the brain; also the delivery of curcumin in the cytoplasm may alter its effect.
- The section on diagnostic tool needs to be updated and account for the recent work.
- Could the authors mention that the flies reported in reference 67 are Drosophila (page 5, lines 203-204)? Could the authors comment on the use of Drosophila as an animal model of AD?
- In the therapeutic tool section, the authors wrote that: ”Unfortunately, AD preclinical research is plagued by poor study design including a lack of blinding and randomisation and no sample size calculation [71,72] despite calls for these controls [73].” The references used seem to be outdated. Could the authors justify for these references or provide more recent ones?
- In the therapeutic tool section, could the authors provide a table that will regroup all the studies using curcumin nanoparticles to treat neurodegenerative studies (as mentioned in point 9)? Please explain why these studies were chosen.
- The section on therapeutic tool is only reporting the use of curcumin nanoparticles from the treatment of AD. Could the authors explain this choice while the use of curcumin nanoparticles has been assessed for other neurodegenerative diseases in recent publications?
- The authors mentioned that “Obviously, the consumption of high doses of Curcumin in the Indian population might be just one of the reasons behind the huge difference in the prevalence of dementia, and several other genetic or behavioural factors are likely to have an important role”. Could the authors provide a reference for this statement?
- Could the authors mention the clinical trials that assessed the effect of curcumin on dementia? The list could be found on clinical trial.org.
Reviewer 2 Report
This review by Gagliardi et al attempts to summarize the findings of in vitro, pre-clinical as well as clinical studies designed to examine the beneficial role of curcumin in neurological diseases. Although the authors have cited most of the relevant studies and made a strong case for future exploration of curcumin as a potential therapeutic option, several critical issues related to curcumin have not been addressed. One of the major issues with curcumin supplementation in vivo is related to its bioavailability; significant mechanistic effects and underlying pathways are described using in vitro studies but comparable effective concentrations of curcumin are rarely detected in vivo. Consequently, tremendous efforts are directed towards new formulations as summarized in this article. However, recent advancements have clearly established the following:
- Intestine-specific effects of curcumin where it significantly improves intestinal barrier function
- Role of unhealthy gut or leaky gut in the development of neurological disorders
Unfortunately, the authors fail to acknowledge the extensive research related to these two aspects which is likely to provide the most significant connection between the effectiveness of curcumin in neurological diseases. The authors recognize the importance of inflammation in the process of development of these diseases but, despite the established role of leaky gut in chronic inflammation, do not adequately incorporate these new concepts.
In addition, the central issue of limited bioavailability underscores the importance of carefully reviewing the published studies in the context of route of administration and levels achieved at the site of action and how do these levels compare with those used to demonstrate underlying mechanisms using in vitro cell culture-based systems.
Overall, the authors summarize the selected studies but the lack of incorporation of new concepts significantly reduces the impact and novelty of this review article.
Author Response
Dear reviewer,
Thanks for your suggestion; we agree that we have missed the aspect about gut absorption and the importance in neurological diseases. We have added this aspect about curcumin intestine absorption and its role in neurological diseases.
We have also integrated the manuscript with a table that summarize the obtained results about curcumin and neurological diseases.
The manuscript has been revised again by Dr. Miriam Hickey, co-author of this paper that is native speaking English.
Round 2
Reviewer 1 Report
The authors addressed all the comments from the reviewers.
Author Response
Thanks for your approval.
Best regards
Stella Gagliardi